# Psychological Stress and Hand Eczema in Physicians and Dentists: A Comparison Based on Surgical Work

**DOI:** 10.3390/bs13050379

**Published:** 2023-05-04

**Authors:** Iva Japundžić, Liborija Lugović-Mihić, Adrijana Košćec Bjelajac, Jelena Macan, Ina Novak-Hlebar, Marija Buljan, Mario Zovak, Dinko Vidović, Zlatko Trkanjec, Matea Kuna

**Affiliations:** 1Department of Dermatovenereology, Sestre Milosrdnice University Hospital Center, 10000 Zagreb, Croatia; iva.japundzic@gmail.com (I.J.);; 2School of Dental Medicine, University of Zagreb, 10000 Zagreb, Croatia; 3Occupational Health and Environmental Medicine Unit, Institute for Medical Research and Occupational Health, 10000 Zagreb, Croatia; 4Department of Traumatology, Sestre Milosrdnice University Hospital Center, 10000 Zagreb, Croatia; 5Department of Neurology, Sestre Milosrdnice University Hospital Center, 10000 Zagreb, Croatia

**Keywords:** psychological stress, hand eczema, physicians, dentists, healthcare workers, occupation, skin, contact dermatitis, psychological factors, quality of life

## Abstract

Background: This research looks at the connection between psychological stress and the prevalence of hand eczema (HE) among physicians and dentists (surgeons, non-surgeons). Methods: This cross-sectional field study involved 185 participants: physicians (surgeons, non-surgeons), dentists (surgeons, non-surgeons) and controls. Hand lesions were examined using the Osnabrueck Hand Eczema Severity Index (OHSI), and participants answered the Nordic Occupational Skin Questionnaire (NOSQ) and Perceived Stress Scale (PSS). Patch tests were performed using commercial contact allergens. Results: The estimated prevalence of HE (self-reported) was 43.9% (physicians 44.6%; dentists 43.2%). HE was significantly more reported by surgeons than controls (*p* < 0.004; V = 0.288). Degrees of perceived stress (PSS) did not differ significantly between the groups, though physicians non-surgeons most exhibited high stress (50%), and physicians surgeons most exhibited low stress (22.5%). High stress was associated with 2.5 higher odds for self-reported HE (*p* = 0.008). Low stress was greater among physicians/dentists who did not report eczema (41.0% vs. 24.6%); moderate stress was more common among those who reported eczema (72.3% vs. 51.8%; *p* = 0.038; V = 0.210). Conclusions: Since high stress levels may negatively influence physicians’/dentists’ work and quality of life, measures to decrease stress could be introduced into the treatment of healthcare workers who are prone to it.

## 1. Introduction

In the last 20–30 years, psychological stress, in general, has become very common; it can be caused/triggered by many factors, including occupation [1,2]. Recent research on occupational stress and work engagement revealed significant stress levels in 19.4% of primary healthcare physicians, where the prominent stressors were a lack of prospects for career growth, poor training and distribution of tasks, and insufficient time to perform the job [1]. The study observed that many physicians experienced occupational stress and had high work engagement. Occupational stress both negatively correlated with, and compromised levels of, work engagement and significantly interfered with the physicians’ ability to have a positive relationship with the work environment [1]. Thus, healthcare workers can suffer from occupational stress as a result of various organisational factors, such as long working hours, shift work, a high workload, lack of job training, low levels of social support at work, etc. [3]. Another study found that stress-related factors at the workplace (e.g., working with suffering and dying patients, organisational problems, and conflicts) increase the risk of distress and burnout [4]. According to previous studies, the proportion of health professionals showing stress above threshold levels has stayed remarkably constant at around 28%, regardless of whether the studies were cross-sectional or longitudinal, compared with around 18% in the general working population [5]. Recent research conducted on 426 resident doctors observed a relatively high burnout rate, citing numerous burnout-related factors and sources of stress in the workplace [6]. One study on occupational stress in healthcare workers, predominantly physicians (60%), found that longer working hours, a higher level of education, and having many children contributed to a higher level of stress [7].

Among physicians, the type of job is an important factor in the recurrence and/or persistence of stress [8,9,10,11]. Surgical work, for example, is highly demanding. According to older research data from a British study, surgeons exhibited high incidences of stress, psychiatric illness, and burnout syndrome [10]. A surgeon’s job involves various stressful situations and may be a trigger for, or contribute to, occupational skin lesions such as eczema [11]. Skin inflammation involves many factors known to be influenced by stress, and certain areas of dermatology could offer useful recommendations related to surgeons’ work, especially concerning the hands and surgery itself (requiring optimization before and after surgery) [8,9,11].

Physicians and dentists are prone to the development/occurrence of skin diseases such as occupational contact dermatitis (CD), which may be influenced by psychological stress. Namely, skin inflammation involves many factors known to be influenced by stress, including the interaction of skin immune cells, hormones, and neurotransmitters [8,9]. Hand eczema (HE) is a commonly reported/recorded occupational dermatosis among healthcare workers. Eczema (from the Greek “ēkzema”, “to boil over”) is a form of dermatitis, or inflammation, of the epidermis [12,13]. Hand dermatitis, a common acute or chronic eczematous disorder, affects the dorsal and palmar sides of the hands and can have a variety of causes [14]. For healthcare workers, a key aetiological factor can be psychological stress (considering the molecular basis of eczema and its link to neuropsychiatric conditions). Since HE impairs both working ability and quality of life, and chronic HE can carry a high economic burden, analysing the relationship between psychological stress and HE may help us to better understand how to manage the disease in healthcare workers and avoid such costs [15,16].

The purpose of this study was to determine the connection between psychological stress and the prevalence of hand eczema among physicians and dentists, both surgeons and non-surgeons (compared to persons not occupationally exposed to skin irritants/allergens), as well as to determine the association between psychological stress and hand eczema confirmed by objective and subjective assessments.

## 2. Materials and Methods

### 2.1. Participants

This cross-sectional epidemiological field study involved 185 participants divided into 5 groups of 37 subjects each: (1) physicians surgeons; (2) physicians of non-surgical professions; (3) dentists surgeons; (4) dentists of non-surgical professions; (5) a control group of employed adults without professional exposure to skin irritants and/or allergens, some of whom were psychotherapists and administrative workers (Figure 1). The research was conducted during the period between March 2018 and April 2019. The participants (physicians, dentists, controls) were drawn from several health care institutions (Sestre Milosrdnice University Hospital Center, Zagreb, Croatia (84 participants); the School of Dental Medicine, University of Zagreb, Zagreb, Croatia (61 participants); University Hospital Dubrava, Zagreb, Croatia (8 participants); the Dental Outpatient Clinic, Zagreb, Croatia; (5 participants); and the Institute of Medical Research and Occupational Health, Zagreb, Croatia (27 participants). It is important to note that different physicians/dentists use different types of gloves in their practice (latex, nitrile and vinyl gloves, depending on preference and availability).

Each participant provided a written consent for inclusion in the study, and the study was approved by each institution’s ethical committee, following the principles of good clinical practice (protocol code EP-15006/17-3).

### 2.2. Methods

#### 2.2.1. Clinical Dermatological Examination, OHSI and NOSQ, and Patch Test

All participants were physically examined by a dermatology specialist, who recorded findings specifically concerning the skin of the hands [17,18]. The presence and severity of changes/lesions on the hands were assessed by the Osnabrueck Hand Eczema Severity Index (OHSI) [19].

Participants were also asked about the manifestations of their occupational skin diseases and exposure to irritants/allergens by means of a modified Nordic Occupational Skin Questionnaire (NOSQ) with questions added about surgical work, years of professional work, and skin dryness [20] (Table 1).

We asked the participants who manifested hand eczema to undergo patch testing. Thus, patch tests were performed (to uncover potential contact allergies) using commercial contact allergens (rubber additives, acrylates, and methylisothiazolinone/methylchloroisothiazolinone) (Chemotechnique Diagnostics, Vellinge, Sweden) [21,22].

#### 2.2.2. Measurement of Stress Level

The participants’ stress levels were measured using the Perceived Stress Scale (PSS) [23], a widely used tool that measures respondents’ perceptions of stressful elements in their life, with questions specifically designed to determine the individual’s sense of control over their disease, emotions related to their disease, and how unpredictable they feel that life is. The PSS has a recall period of one month, and answers are given on a Likert scale, where 0 is “never” and 4 is “very often”. The higher the sum of their scores, the greater the respondent’s perceived stress, the severity of which is categorised as low, moderate, or high. The test exhibits good reliability (0.84–0.86).

#### 2.2.3. Statistical Analysis

The prevalence of hand eczema was assessed using a confidence interval (CI) [24]. A 95% CI was determined using an Internet calculator and the Wilson procedure without corrections [25]. Overlapping between the objective and subjective findings for hand eczema was measured using Cohen’s kappa coefficient [26,27].

For the comparison of the proportions, the Fisher’s exact test and χ2 test were used, as well as the post-hoc *z*-test with the Bonferroni correction for multiple comparisons. The effect size, as a measure of the size of the difference between the groups, was quantified with Cramer V. The interpretation was performed using the Cohen criteria: 0.1–0.3 = a small effect size; 0.3–0.5 = a medium effect size; 0.5–0.7 = a large effect size; and >0.7 = a very large effect size. Odds ratios with 95% confidence intervals (CI) were calculated for the relationship between the stress level and eczema. IBM SPSS 22 software (IBM Corp, Armonk, NY, USA) was used for the statistical analysis, and *p* values lower than 0.05 were considered statistically significant.

## 3. Results

Our study included 185 participants (60% women and 40% men), varying in age between 26 and 76 (median 41).

Out of 69 healthcare workers with hand eczema, 34 agreed to be tested using a patch test. The patch test results for potential allergies predominantly indicated non-allergic eczema (i.e., irritant CD) (91.2% (95% CI 77.0–97.0%)). Only a few results confirmed contact allergies (i.e., allergic CD) (8.82% (95% CI 3.0–23.0%)). Thus, positive patch test results (one or more positive reactions) were recorded only to acrylates in three dentists non-surgeons.

Although the degrees of perceived stress (PSS) did not differ significantly between the five groups, some observations/differences were recorded (Table 2). An analysis of the psychological stress levels by group shows that high stress was most commonly recorded among physicians non-surgeons (50%), followed by dentists surgeons (25%), and finally, physicians surgeons and dentists non-surgeons (12.5%); none of the controls reported high stress (0%) (Table 2). Low stress was most commonly recorded among the controls (29.6%), followed by physician surgeons (22.5%), dentists surgeons (18.3%), and dentists non-surgeons and physicians non-surgeons (16.9% each).

We examined both the participants’ self-reports and the data recorded at the dermatological examinations. The dermatological findings and self-reported histories of previous hand eczema were analysed separately.

### 3.1. Results Based on Self-Reported Histories of Previous Hand Eczema and Data on Participants’ Stress Levels

Based on the self-reports for hand eczema, the estimated prevalence of hand eczema was 43.9% (95% CI 36.2–52.0%). By group, the prevalence was 44.6% (95% CI 33.8–55.9%) for physicians (surgeons and non-surgeons) and 43.2% (95% CI 32.6–54.6%) for dentists (surgeons and non-surgeons). Hand eczema was significantly more often reported by physicians/dentists of surgical professions than by controls, with a low effect size (*p* < 0.004; V = 0.288) (Table 3). However, concerning the data on self-reported eczema in relation to surgical professions, there was no significant difference between the physicians and dentists, or any difference between the surgical and non-surgical professions. In addition, the physicians/dentists of non-surgical professions did not differ from those of the surgical professions or from the controls (Table 3).

By stress level, the participants who reported hand eczema (group “Eczema+”, *N* = 80), compared to those who did not report hand eczema (group “Eczema−“, *N* = 105), had more moderate stress and less low stress (Table 4). When the stress level was dichotomised (0 = low stress vs. 1 = moderate + high stress), moderate or high stress was associated with 2.5 higher odds for self-reported hand eczema (95% CI 1.3–4.7; *p* = 0.008).

A low level of perceived stress was significantly more often recorded among physicians/dentists who did not report eczema than among those who reported eczema (41.0 vs. 24.6%), while moderate stress was more common among those with reported eczema than those without eczema (72.3 vs. 51.8%; *p* = 0.038; V = 0.210; Table 5).

### 3.2. Objective Dermatological Data Recorded (During Examination) for Hand Eczema in Physicians/Dentists and Controls and Data on Psychological Stress Levels

When taking into account the objective skin findings (skin changes/lesions) observed during a dermatological examination, the prevalence of hand eczema was 45.3% (67/148; 95% CI 37.5–53.3%) (Table 6).

Although high stress levels were observed in some physicians/dentists, our results did not reveal a significant association between increased stress levels and objectively established hand eczema (at the moment of examination) for any of the groups (Table 7 and Table 8).

When analysing stress levels in relation to objectively determined hand eczema, subjects with objectively diagnosed hand eczema (eczema+) did not differ significantly by degrees of perceived stress from the subjects without eczema (eczema−) (Table 7).

When analysing levels of perceived stress (PSS) in comparison to objectively/dermatologically determined hand eczema among physicians/dentists (with present (*N* = 67) and absent (*N* = 81) eczema (eczema+; eczema−)), the perceived stress levels did not differ between those with objectively determined eczema from those without eczema (Table 8). Moderate or high stress was not associated with objective eczema, neither for the total sample nor between the four groups of physicians/dentists.

When comparing disease severity (OHSI) and stress, we found no relationship between the OHSI results and stress levels, neither for the total sample nor for the separate groups (according to the Spearman’s correlation and Kruskal–Wallis tests) (Figure 2).

There was no significant difference between the three groups with different stress levels concerning the duration of work as a doctor and the hours of work per week. The same stress levels were also observed among those who performed additional work and those who did not (30.7 vs. 22.5%). When dichotomising the duration of work as a doctor, hours of work per week and stress level (using logistics regression), none of these variables were a significant predictor of stress.

Between the five groups, the only significant difference found was for skin desquamation, which was most common in physicians surgeons (Table 9 and Table 10).

## 4. Discussion

### 4.1. Our Key Findings on Psychological Stress and Eczema in Comparison to the Literature Data

The results of our field study, and the reports of our participants (physicians’ and dentists’) on their hand eczema, support an association between psychological stress and hand eczema. This may indicate that work-related psychological stress in physicians and dentists may be associated with their reports on their hand eczema, i.e., their history of previous manifestations of hand eczema. However, dermatological/clinical findings of hand eczema in our examinees were not associated with an increased level of stress, which is understandable because of the temporary nature of their skin lesions/changes.

When looking at the literature data, generally, there are contradictory results on the association between the occurrence of hand eczema (and CD) and psychological stress [28]. Some studies support a positive association/correlation, while others did not observe this association [29,30,31]. The relationship between hand eczema severity and stress level was examined in a newer study with 109 participants using an assessment of hand eczema severity (the OHSI) and the Perceived Stress Scale-10 (PSS-10) [28]. The results showed high perceived stress in 18.3% of examinees and a significant association between their hand eczema severity and the high level of stress [28]. Similarly, based on the statistical analysis of our collected data for physicians/dentists, a moderate or high level of stress was associated with 2.5 higher odds of reports of hand eczema (*p* = 0.008). This may confirm the influence of occupational stress on the condition of the skin.

In addition, according to the research literature, psychological stress may activate or exacerbate dermatoses, e.g., atopic dermatitis (AD)—this relationship is important because hand eczema is common in patients with AD. According to a recent, large study (57,046 participants)*,* hand eczema occurrence during the past year was positively associated with chronic stress, similar to the findings of previous studies by Anveden et al. and Hamnerius et al. [30,32,33,34]. It was shown that having hand eczema, especially a severe form, can cause stress for patients. However, in a recent study by Loman, results indicate that stress contributes to the occurrence of hand eczema rather than hand eczema causing stress [33].

### 4.2. Relationship between Occupation, Psychological Factors, Lifestyle, and Hand Eczema

For occupational hand eczema, however, there is no clear confirmation of a relationship between chronic stress, job-related stress and sickness absence [35]. According to one study that analysed 122 patients with occupational hand eczema for a possible association between chronic stress (or burnout symptoms) and cumulative sickness absence, occupational hand eczema manifestations were not more severe among those who experienced greater stress and burnout. For some measures, higher chronic stress levels were reported by women, and cumulative days of sickness absence correlated with individual dimensions of job-related stress. This suggests that for patients with severe occupational hand eczema, chronic stress is an additional factor predicting cumulative sickness absence [35]. According to another study on stress levels (assessed by PSS) in patients with hand eczema, significant stress levels were frequently recorded (67.7% of patients), but with no significant correlation between clinical features and specific aetiologies [36].

In addition to stress, other lifestyle factors can potentially influence hand eczema. Recently, the association between lifestyle factors (smoking, alcohol consumption, physical activity, diet, body mass index, and sleep) and the occurrence/features of hand eczema was examined by a systematic review and meta-analysis [33]. While it did find a small amount of evidence for a relative association between smoking and a higher hand eczema prevalence, it found no convincing evidence of associations to the other lifestyle factors [33]. Concerning the relationship between hand eczema regression and stress, in one study involving 1491 patients, complete regression at follow-up was reported by 19.3% of the patients—having a high level of stress and current tobacco use (smoking) were significant factors that negatively affected the regression/healing of hand eczema, while a high level of exercise was significantly related to regression/healing [37].

Magnavita et al. analysed the skin lesions and psychosocial factors at work in 1744 hospital workers and showed that the workers’ skin problems were not supported by objective evidence and that many patients only had mild skin disease [29]. In addition, their personality characteristics (e.g., neuroticism and negative moods) may have influenced the perception of stressors at work and led to the overestimation of skin lesion severity [29]. Our results can be interpreted in a similar way, since the only association found was between the self-reported moderate stress levels and self-reported eczema symptoms. In addition, our observation that individuals who experienced moderately high stress levels are more prone to report hand eczema symptoms is probably related to psychological factors. Thus, we were the first to include groups of physicians and dentists by surgery job and used specific questionnaires to examine a relationship between occupation, psychological stress and hand eczema.

### 4.3. Surgical Work, Psychological Stress and Hand Eczema

It is important to note that, according to our results, hand eczema was significantly more commonly reported by the surgeons (physicians and dentists) than by the controls (with a low effect size). These results indicate that our groups of surgeons did not experience such high levels of stress, a welcome sign since stress can negatively affect surgeons’ performance during surgical procedures, jeopardizing patient safety [38]. According to one study in which stress levels in surgeons and residents was measured using a smart patch, the highest level of stress was associated with performing an operation, particularly in fellows and residents [38]. In another study on surgeons, trainees with the highest burnout rate also had a higher rate of moderate or severe depression, higher perceived stress, and lower social support and self-efficacy, among other factors [2]. In other words, higher perceived stress and higher depression were associated with higher burnout scores, while lower perceived stress was associated with lower burnout scores [2].

Although the degree of perceived stress did not significantly differ between our groups, a lower level of stress was most often observed in surgeons (along with members of the control group), while high stress was most often observed in non-surgeons. Therefore, these results support previous authors’ assumptions that surgeons are a special and homogeneous group with specific temperament and personality traits, and cope well with stress [39,40]. Surgeons very often find themselves in stressful situations at work, especially taking into account the environment of modern surgical practice, burdened with administrative, technological and interpersonal factors. Stressors can be temporary, but the repetition of stressful situations can lead to burnout [41,42,43,44]. In addition, more recent studies have noted the increasing incidence of burnout syndrome at work among doctors, especially surgeons, with the prevalence of burnout among physicians between 29% and 55%, and its rise among surgeons [43]. In addition, surgeons show a high level of cynicism and burnout due to exhaustion, regardless of their narrow specialty or weekly working hours [45,46]. Furthermore, in assessing the impact of work on surgeons, the damage that can result from long-term work in poor conditions is rarely taken into account, so even when symptoms of discomfort appear, they are often ignored. As surgeons navigate a high-productivity, fast-paced environment, thoughts/complaints related to stress and fatigue are often suppressed or ignored [47].

### 4.4. Current Knowledge and Findings on Psychoneuroimmune Aspects of Eczema

Although several experimental studies have pointed to the connection between stress and impaired function of the epidermal barrier, there is little literature on the connection between stress and the clinical appearance and severity of skin lesions [28,48,49]. However, there is knowledge that stress weakens the functioning of the epidermal barrier and inhibits its recovery, which implies a link between stress and the worsening or long-term regression/healing of skin diseases [49]. A possible explanation for the positive association between the occurrence/severity of hand eczema and stress may come from the field of psychoneuroimmunology and the link between the central nervous system (CNS) and the immune system via the hypothalamic–pituitary–adrenal (HPA) axis [8,30]. In addition, when considering an explanation for the mechanism by which stress influences the occurrence of skin lesions, the various psychoneuroimmune factors and the skin’s very complex network, need to be taken into account. This, then, involves the field of psychoneuroimmunology, as mentioned in the literature.

The body responds to stress by activating the HPA axis, which regulates the release of adrenocorticotropic hormone (ACTH), β-endorphins, and cortisol, and by activating the sympatho-adrenal medullary system, which regulates the release of adrenaline and noradrenaline [50]. Glucocorticoids reduce the production of lipids, and thus the production and excretion of lamellar bodies. Consequently, the production of lamellar membranes is hindered in the stratum corneum, weakening the barrier function of the stratum corneum [51]. Psychological stress might lead to the release of corticotropic-releasing hormone (CRH), which activates the HPA axis, leading to increased endogenous glucocorticoid, a modulation of the inflammatory response, and a decrease in epidermal lipid synthesis, antimicrobial defence, and barrier ability [52]. In addition, chronic stress may induce an imbalance of T helper (Th) Th1/Th2 responses in favour of the Th2-mediated response, which can contribute to hand eczema. Another possible reason for the association between hand eczema and psychological stress for some patients with hand eczema could be their atopy. One of the main factors associated with hand eczema is atopy—atopic diseases such as AD can be triggered by stress as well [30,53,54,55]. In addition, according to research data, skin acts as a local analogue of the hypothalamic–pituitary–adrenal axis—it expresses elements of that axis, which include proopiomelanocortin (POMC), corticotropin-releasing hormone (CRH), CRH receptor-1 (CRH-R1), and key enzymes for corticosteroid and glucocorticoid synthesis [8,56]. In addition, in the epidermis there are free nerve endings that connect the skin to the peripheral nervous system and release neuropeptides during stress (e.g., substance P and vasoactive intestinal peptide (VIP)) that affect the mitotic activity of keratinocytes. In addition, the nerve endings themselves can directly affect the activity of Langerhans cells and, thus, contribute to the onset or worsening of skin diseases during stress [48]. All these processes are very complex, and further studies are welcome.

### 4.5. Implications, Strengths and Limitations of the Study

To gain better insight into the connection between psychological stress and the prevalence/occurrence of hand eczema in physicians and dentists, various factors must be taken into account. The results of one study found that certain factors (high stress levels, smoking and exercise) were important to the prognosis of occupational hand eczema, while some common risk factors did not always affect prognosis (AD and contact sensitivity) [28]. Another consideration is that more than half of our physicians/dentists reported skin lesions on the hands; thus, we support other authors’ conclusions that promoting preventive measures as a lifestyle is useful. Positive stress management, for example, could improve physicians’ and surgeons’ work-life balance, minimizing the impact of stressors in the workplace and the severity of their hand eczema.

In addition to healthcare professionals’ high exposure to stress, they are also immersed in the medical field’s culture of perfectionism. They need tools to achieve a balance between the expectations of work and the need for rest in order to help their patients effectively [57]. According to the literature data, the most common and efficient coping strategies include social and emotional support, physical activity, physical self-care, and emotional and physical distancing from work [58]. According to one study, appropriate coping strategies could be employed among co-workers for the prevention of psychological suffering, especially when working in stressful conditions [58]. Previous research/evaluations of coping strategies for the treatment of dermatoses (meditation, biofeedback, hypnosis, guided imagery, and others) revealed beneficial effects concerning itching, psychosocial outcomes, and even skin severity [59]. For example, mindfulness meditation may alleviate stress, depression, and burnout, and strengthen well-being and empathy among doctors and other healthcare workers, improving the quality of care they can provide patients. In turn, this can potentially contribute to a wider, positive effect on healthcare services in general [57].

One strength of this study is that it is the first study to look at the connections between psychological stress and the prevalence of hand eczema among physicians and dentists of the non-surgical and surgical professions (in relation to healthy/control persons not occupationally exposed to skin irritants/allergens). In addition, our study was the first to look at the possible association between psychological stress and hand eczema, which had been confirmed by both objective and subjective factors. In addition, this was the first study to use multiple questionnaires on hand eczema and occupation for the same sets of examinees (physicians/dentists), and in which stress was analysed and compared in physicians and dentists in the same study.

Concerning the limitations of the study; there was a small number of participants—further studies with greater numbers of participants would provide useful data. In addition, it is necessary to mention that many other variables could possibly influence the occurrence of hand eczema in physicians/dentists aside from psychological stress and exposure to skin irritants/allergens, such as the frequency of handwashing, types of gloves used, gender, age, lifestyle, and place of work, among others [17,18]. It should also be noted that participants from the same group have different habits, e.g., doctors use gloves made of different materials (latex, nitrile or vinyl; with talc or without talc) depending on personal preferences.

## 5. Conclusions

Our findings support an association between psychological stress and eczema—among physicians/dentists who reported eczema, moderate stress was significantly more common, while among physicians/dentists who did not report eczema, low stress levels predominated. Some physicians and dentists experience high psychological stress related to their hand eczema, and it is necessary to take this into account when analysing their health status and occupational tasks. Since almost every second doctor (physicians and dentists) in our study had skin changes/lesions on the hands, which may impair their work and limit their daily activities, special attention should be paid to designing and implementing preventive measures for them and other healthcare workers. In addition, since high stress levels may negatively influence physicians/dentists’ work and their quality of life, additional measures to decrease stress could be involved/introduced to the treatment of healthcare workers who are prone to it. Understanding the aetiological factors of stress and their effect on quality of life are very valuable for the management of stress itself and, therefore, hand eczema.

## Figures and Tables

**Figure 1 behavsci-13-00379-f001:**
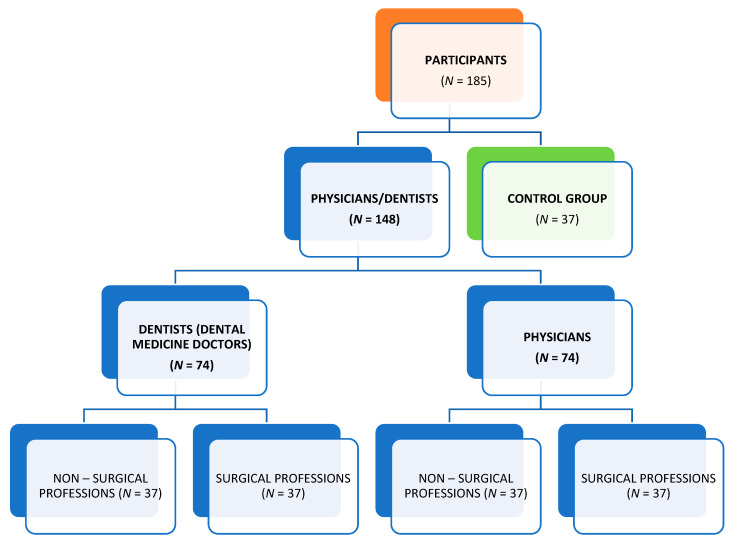
Groups of participants involved in the study.

**Figure 2 behavsci-13-00379-f002:**
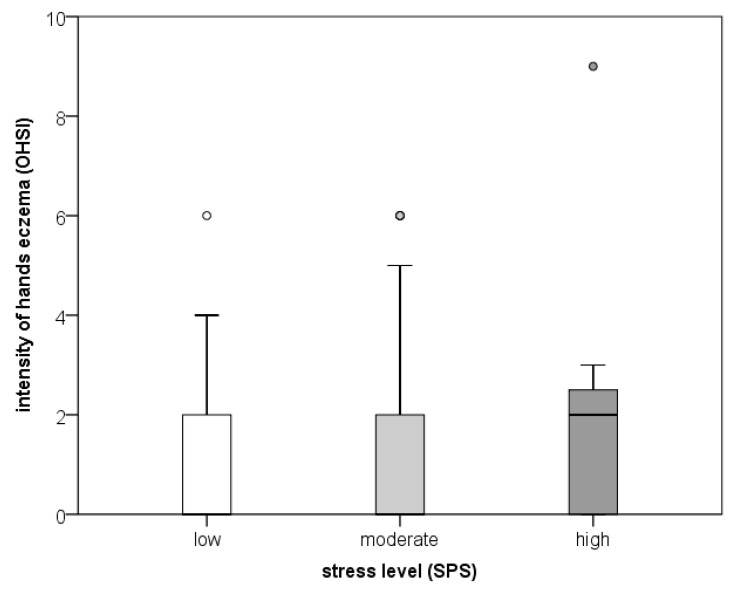
OHSI values by stress level. Circles represent outliers.

**Table 1 behavsci-13-00379-t001:** The study questionnaire (modified Nordic Occupational Skin Questionnaire, NOSQ).

Factor	Question	Answer
Present Occupation	Workplace (physician/dentist)	_________________________
Surgical Work	Do you usually perform/do surgical work?	_________________________
Age	Year of birth	_________________________
Gender	Man/woman	_________________________
Main Activity at Work	What is your main activity at work as a physician/dentist?	_________________________
Duration of Work as a Doctor	How long have you been working as a physician/dentist?	_________________________
Hours of Work per Week	How many hours per week do you work in your main job (on average)? hours/week	_________________________
Additional Job	Do you perform any other paid job regularly?	_________________________
Skin Dryness	Do you have dry skin?	_________________________

**Table 2 behavsci-13-00379-t002:** Perceived stress levels (PSS) by group of participants (*N* = 185; 37 respondents per group).

Group								*p **	V
Variable			Physicians Non-Surgeons (*N* = 37)	Physicians Surgeons (*N* = 37)	Dentists Non-Surgeons (*N* = 37)	Dentists Surgeons (*N* = 37)	Control Group (*N* = 37)		
Stress	Low	*N*/%	9 (12.7%)	16 (22.5%)	12 (16.9 %)	13 (8.3%)	21 (29.6%)		

	Moderate	*N*/%	24 (22.6%)	20 (18.9%)	24 (22.6%)	22 (20.8%)	16 (15.1%)		

	High	*N*/%	4 (50%)	1 (12.5%)	1 (12.5%)	2 (25.0%)	0 (0.0%)	0.090	0.192

* χ^2^ test.

**Table 3 behavsci-13-00379-t003:** Prevalence of hand eczema based on self-reported data of previous hand eczema in each group of participants.

	Group					*p* *	V
	Physicians Non-Surgeons (*N* = 37)	Physicians Surgeons (*N* = 37)	Dentists Non-Surgeons (*N* = 37)	Dentists Surgeons (*N* = 37)	Controls (*N* = 37)		
*N*/%	13 ^ab^ (35.1%)	20 ^b^ (54.1%)	14 ^ab^ (37.8%)	18 ^b^ (48.6%)	5 ^a^ (13.5%)	**0.004**	0.288

* χ^2^ test. ^ab^ Groups sharing the same superscript letter did not differ significantly based on the *z*-test for proportions with Bonferroni corrections for multiple comparisons. The bold represents significant differences.

**Table 4 behavsci-13-00379-t004:** Levels of perceived stress among participants who reported hand eczema (*N* = 80) and those who did not report hand eczema (*N* = 105).

Variable			Eczema− (*N* = 105)	Eczema+ (*N* = 80)	Total	*p **	V
Stress	Low	*N*	53 ^a^ (46.1%)	18 ^b^ (25.7%)	71 (38.4%)		

	Moderate	*N*	56 ^a^ (48.7%)	50 ^b^ (71.4%)	106 (57.3%)		

	High	*N*	6 ^a^ (5.2%)	2 ^a^ (2.9 %)	8 (4.3%)	**0.010**	0.223

* χ^2^ test. ^ab^ Groups sharing the same superscript letter did not differ significantly based on the *z*-test for proportions with Bonferroni corrections for multiple comparisons. The bold represents significant differences.

**Table 5 behavsci-13-00379-t005:** Levels of stress among physicians/dentists who reported having eczema (eczema+) (*N* = 65) and those who did not report hand eczema (eczema−) (*N* = 83).

Variable			Eczema− (*N* = 105)	Eczema+ (*N* = 80)	Total (*N* = 185)	*p **	V
Stress	Low	*N*	34 ^a^ (41.0%)	16 ^b^ (24.6%)	50 (33.8%)		

	Moderate	*N*	43 ^a^ (51.8%)	47 ^b^ (72.3%)	90 (60.8%)		

	High	*N*	6 ^a^ (7.2%)	2 ^a^ (3.1%)	8 (5.4%)	0.038	0.210

* χ^2^ test. ^ab^ Groups sharing the same superscript letter did not differ significantly based on the *z*-test for proportions with Bonferroni corrections for multiple comparisons.

**Table 6 behavsci-13-00379-t006:** Objective data from a clinical examination of the skin of the hand (OHSI indicators) for all participants (*N* = 185) (by group) with a comparison of the prevalence of hand eczema.

	Group					*p* *	V
	Physicians Non-Surgeons (*N* = 37)	Physicians Surgeons (*N* = 37)	Dentists Non-Surgeons (*N*= 37)	Dentists Surgeons (*N* = 37)	Control Group (*N* = 37)		
*N*/%	14 (37.8%)	15 (40.5%)	21 (56.8%)	17 (45.9%)	13 (35.1%)	0.354	0.154

* χ^2^ test.

**Table 7 behavsci-13-00379-t007:** Levels of perceived stress (PSS) in all participants with present (*N* = 80) and absent (objectively determined) hand eczema (*N* = 105) (measured by OHSI).

Variable			Eczema− (*N* = 80)	Eczema+ (*N* = 105)	Total (*N* = 185)	*p*	V
Stress *	Low	*N*	44 (41.9%)	27 (33.8%)	71 (38.4 %)		
	Moderate	*N*	58 (55.2%)	48 (60.0%)	106 (57.3%)		
	High	*N*	3 (2.9%)	5 (6.3%)	8 (4.3%)	0.337	0.108

* χ^2^ test.

**Table 8 behavsci-13-00379-t008:** Levels of perceived stress (PSS) among physicians/dentists with present (*N* = 67) and absent (*N* = 81) (objectively determined) hand eczema (eczema+; eczema−) (measured by OHSI).

Variable			Eczema− (*N* = 80)	Eczema+ (*N* = 105)	Total (*N* = 185)	*p*	V
Stress *	Low	*N*	28 (34.6%)	22 (32.8%)	50 (33.8%)		
	Moderate	*N*	50 (61.7%)	40 (59.7%)	90 (60.8%)		
	High	*N*	3 (3.7%)	5 (7.5%)	8 (5.4%)	0.602	0.083

* χ^2^ test.

**Table 9 behavsci-13-00379-t009:** Values for components of the OHSI and dry skin by study group.

		Physicians Non-Surgeons (*N* = 37)	Physicians Surgeons (*N* = 37)	Dentists Non-Surgeons (*N* = 37)	Dentists Surgeons (*N* = 37)	Control Group (*N* = 37)	*p* *	V
Erythema	*N*/%	10 (27.0%)	12 (32.4%)	17 (45.9%)	8 (21.6%)	13 (35.1%)	0.225	0.175
Desquamation	*N*/%	4 ^a^ (10.8%)	5 ^b^ (13.5%)	1 ^a^ (2.7%)	0 ^a^ (0.0%)	0 ^a^ (0.0%)	**0.020**	0.251
Papules	*N*/%	1 (2.7%)	0 (0.0%)	2 (5.4%)	0 (0.0%)	2 (8.1%)	0.210	0.178
Induration	*N*/%	1 (2.7%)	7 (18.9%)	6 (16.2%)	8 (21.6%)	7 (18.9%)	0.178	0.185
Fissures	*N*/%	1 (2.7%)	0 (0.0%)	2 (5.4%)	4 (10.8%)	0 (0.0%)	0.081	0.212
Dry Skin	*N*/%	6 (16.2%)	5 (13.5%)	13 (35.1%)	6 (16.2%)	5 (13.5%)	0.088	0.209

* χ^2^ test. ^ab^ Groups sharing the same superscript letter did not differ significantly based on the *z*-test for proportions with Bonferroni corrections for multiple comparisons. The bold represents significant differences.

**Table 10 behavsci-13-00379-t010:** Percentages for components of the OHSI and dry skin by study group.

		Group					*p* *	V
		Physicians Non-Surgeons	Physicians Surgeons	Dentists Non-Surgeons	Dentists Surgeons	Control Group		
Erythema	*N*	10	12	17	8	13		
	%	16.7%	20.0%	28.3%	13.3%	21.7%	0.225	0.175
Desquamations	*N*	4 ^a^	5 ^b^	1 ^a^	0 ^a^	0 ^a^		
	%	40.0%	50.0%	10.0%	0.0%	0.0%	**0.020**	0.251
Papules	*N*	1	0	2	0	2		
	%	16.7%	0.0%	33.3%	0.0%	50.0%	0.210	0.178
Induration	*N*	1	7	6	8	7		
	%	3.4%	24.1%	20.7%	27.6%	24.1%	0.178	0.185
Fissures	*N*	1	0	2	4	0		
	%	14.3%	0.0%	28.6%	57.1%	0.0%	0.081	0.212
Dry Skin	*N*	6	5	13	6	5		
	%	17.1%	14.3%	37.1%	17.1%	14.3%	0.088	0.209

* χ^2^ test. ^ab^ Groups sharing the same superscript letter did not differ significantly based on the *z*-test for proportions with Bonferroni corrections for multiple comparisons. The bold represents significant differences.

## Data Availability

The data presented in this study are available on request from the corresponding author.

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
