# Peer review of "Psychological Stress and Hand Eczema in Physicians and Dentists: A Comparison Based on Surgical Work"

_behavsci, 2023, doi:10.3390/bs13050379_

Round 1

Reviewer 1 Report

The title is relevant and informative

The abstract matches the whole text. It indicates the main topics, results, and the research question is for me clearly outlined. 

The introduction is concise and good but needs some updated references.

Please provide a definition of eczema (and support it with a reference); 

The author has described the aim of the study at the end of the introduction. 

Ethics: The protocol of this study was approved by Ethics Committee as stated:

“Each participant provided a written consent for inclusion in the study, and the study was approved by each institution's ethical committee, following the principles of good clinical practice (Zagreb, EP-15006/17-3)”.

You stated, “the participants (physicians, dentists, controls) were drawn from several health” Please provide the number of participants from each institution. Please specify the period of the survey in the abstract and methods sections.

The tables are informative and relevant.

The design and statistical analysis described in the methods section is in accordance with the objective of the study. The intended sample size and sampling strategy of this study are clear.

Please attach the questionnaire you used.

The results are well described.

It would be worth identifying what are the main implications of this study. Please highlight well the new knowledge generated in this survey and for the examinees.

You stated a limitation of the study: number of participants; I suggest adding a limitation subsection and providing more information regarding the number of participants, the study area, the gender, the age…

Author Response

RESPONSE

Dear Editor and Reviewers,

thank You for Your valuable time, effort and useful contribution You have put into assesing our previous version of our manuscript. We really appreciate the input You have given because it has definitely improved our manuscript.

We have given careful consideration to each comment as follows:

REVIEWER 1

  • The introduction is concise and good but needs some updated references.

We somewhat changed the text and added some additional references and removed one old reference, according to the reviewer 's suggestion. Thank You!

  • Please provide a definition of eczema (and support it with a reference); 

We added a definition of ezema and more data on „eczema“ and „hand eczema“, according to the reviewer 's suggestion.

  • You stated, “the participants (physicians, dentists, controls) were drawn from several health..” Please provide the number of participants from each institution. Please specify the period of the survey in the abstract and methods sections.

We added the number of participants from each institution, as well as the period of the survey. according to the reviewer 's suggestion. Thank You!

  • Please attach the questionnaire you used.

We added our questionnaire in one table (Table 1) - according to the reviewer 's suggestion.

  • It would be worth identifying what are the main implications of this study. Please highlight well the new knowledge generated in this survey and for the examinees.

We added this data (the main implications of this study) according to the reviewer 's suggestion.

  • You stated a limitation of the study: number of participants; I suggest adding a limitation subsection and providing more information regarding the number of participants, the study area, the gender, the age…

We added more data on limitations of the study in a new paragraph - according to the reviewer 's suggestion. Thank You!

Best regards,

Authors

Reviewer 2 Report

This paper is an excellent and important addition to the literature in the field of Behavioral Science.  Psychological Stress in health professionals and other diseases have a negative influence on healthcare professionals. This cross-sectional field study involved 185 participants and has a small sample size, but valuable because of its data analysis. Eczema is a common disease with economic and social ramifications, especially in stressful settings like those described in this paper.  Previous studies on the occupational health of hospitals show different degrees of phycological stress because of different skin problems, especially eczemas. There has been no comparative study of the effects of various work stressors in different specialties and eczemas.

Additional comments:

In more detail, Japundžić and colleagues show the connection between psychological stress and eczemas in health settings.

The title is a little confusing, it needs to reflect the study. Maybe use a word that groups all those healthcare professionals.

The discussion is too long and hard to read. Maybe it will be better if the authors separate into sub titles.

What are the strengths and limitations of the paper?

What are the practical applications?

The conclusion doesn’t reflect the actual study.

What is the relationship between the stressor and health, and how this is correlated to eczemas?

I don’t see the comparison between specialties or in this case physicians and dentists of surgical and non-surgical professions.

What is the most prevalent symptom correlated with stress levels? Compared to all levels of health professionals described in this paper?

How was stress measured and assessed? What was the main predictor? 

Author Response

RESPONSE

Dear Editor and Reviewers,

thank You for Your valuable time, effort and useful contribution You have put into assesing our previous version of our manuscript. We really appreciate the input You have given because it has definitely improved our manuscript.

We have given careful consideration to each comment as follows:

REVIEWER 2

This paper is an excellent and important addition to the literature in the field of Behavioral Science. In more detail, Japundžić and colleagues show the connection between psychological stress and eczemas in health settings.

-The title is a little confusing, it needs to reflect the study. Maybe use a word that groups all those healthcare professionals.

We modified the title and wrote it to be more specific as much as possible, according to the reviewer 's suggesti on.Thank You!

-The discussion is too long and hard to read. Maybe it will be better if the authors separate into sub titles.

We modified/changed the Discussion and separated it into specific parts under specific subtitles, according to the reviewer 's suggestion.

-What are the strengths and limitations of the paper?

We added more data on the strength and limitations of the study in a separate paragraph, according to the reviewer 's suggestion.Thank You!

-What are the practical applications?

We added more specific data on its practical applications according to the reviewer 's suggestion.

-The conclusion doesn’t reflect the actual study.

We changed the conclusion and added more specific data.Thank You!

-What is the relationship between the stressor and health, and how is this correlated to eczemas?

We conducted additional statistical analysis and we presented additional results in the final part of the Results section, according to the reviewer 's suggestion. Thank You!

-I don’t see the comparison between specialties or in this case physicians and dentists of surgical and non-surgical professions.

We conducted additional statistical analysis – we presented additional results in the final part of the Results section, according to the reviewer 's suggestion.

-What is the most prevalent symptom correlated with stress levels? Compared to all levels of health professionals described in this paper?

We added data obtained by analysis for this association  - as previously mentioned. So, we conducted additional statistical analysis and presented additional results in the final part of the Results section, according to the reviewer 's suggestion.

 - How was stress measured and assessed? What was the main predictor? 

We emphasized PSS as a method for stress measurement and added data obtained by analysis of stress  - as previously mentioned. Also, we conducted additional statistical analysis – we presented additional results in the final part of the Results section, according to the reviewer 's suggestion. Thank You!

Best regards,

Authors

Reviewer 3 Report

There are many variables influencing the occurrence of HE in physicians, not only the possible link with psychological stress or exposure to skin irritants, but also e.g. frequency of hand wash, types of gloves used. Did You take this into consideration?

Author Response

RESPONSE

Dear Editor and Reviewers,

thank You for Your valuable time, effort and useful contribution You have put into assesing our previous version of our manuscript. We really appreciate the input You have given because it has definitely improved our manuscript.

We have given careful consideration to each comment as follows:

REVIEWER 3

-There are many variables influencing the occurrence of HE in physicians, not only the possible link with psychological stress or exposure to skin irritants, but also e.g. frequency of hand wash, types of gloves used. Did You take this into consideration?

We agree with this observation and added potential roles of these factors and their potential influences on HE to the Discussion, according to the reviewer 's suggestion. Thank You!

Best regards,

Authors

Reviewer 4 Report

Thank you for the opportunity to read this interesting article. The design of the study was developed very creatively and is really aimed at studying the problem of the impact of stress and professional burnout on the health of doctors.

For a more complete understanding of the meaning of the article by the reader, I would like to clarify some details. It is obvious that most of the doctors who took part in the survey, with the possible exception of psychotherapists, constantly use gloves. It is known that gloves made of different materials have different purposes and the choice of gloves is determined by the methods used by a particular doctor. In particular, in our practice of genetic research by polymerase chain reaction, we use only nitrile powder-free gloves and no others. Thus, the choice of such an important part of the doctor's equipment in contact with the skin is determined not by his personal preferences and skin condition, but by the methods of work used. This factor (the nature of the glove material and the presence or absence of talc) can have a significant impact on the development of skin diseases. Is it possible that in the country where the study was conducted, certain glove standards apply, for example, latex gloves are not used? It is necessary to clarify this issue when describing research methods. Also, the meaning of the phrases in line 213 that a subjective statement about the presence of a disease could have an impact is not entirely clear. Perhaps it is worth perfrasing this sentence so that its meaning becomes more explicit.

Author Response

RESPONSE

Dear Editor and Reviewers,

thank You for Your valuable time, effort and useful contribution You have put into assesing our previous version of our manuscript. We really appreciate the input You have given because it has definitely improved our manuscript.

We have given careful consideration to each comment as follows:

REVIEWER 4

 - The design of the study was developed very creatively and is really aimed at studying the problem of the impact of stress and professional burnout on the health of doctors. For a more complete understanding of the meaning of the article by the reader, I would like to clarify some details. It is obvious that most of the doctors who took part in the survey, with the possible exception of psychotherapists, constantly use gloves.  It is known that gloves made of different materials have different purposes and the choice of gloves is determined by the methods used by a particular doctor.    In particular, in our practice of genetic research by polymerase chain reaction, we use only nitrile powder-free gloves and no others.  Thus, the choice of such an important part of the doctor's equipment in contact with the skin is determined not by his personal preferences and skin condition, but by the methods of work used.    This factor (the nature of the glove material and the presence or absence of talc) can have a significant impact on the development of skin diseases.    Is it possible that in the country where the study was conducted, certain glove standards apply, for example, latex gloves are not used? It is necessary to clarify this issue when describing research methods.

  We agree with this observation and mentioned all these factors in the text and added potential influences of these factors to the Discussion, according to the reviewer 's suggestion. Thank You!

-Also, the meaning of the phrases in line 213 that a subjective statement about the presence of a disease could have an impact is not entirely clear. Perhaps it is worth perfrasing this sentence so that its meaning becomes more explicit.

We explained this part of the text and refrased it. Thank you!

Best regards,

Authors